# Evolving patterns of COVID-19 mortality in US counties: A longitudinal study of healthcare, socioeconomic, and vaccination associations

**Fardin Ganjkhanloo**[1,2], **Farzin Ahmadi**[1,2], **Ensheng Dong**[1,2], **Felix Parker**[1,2], **Lauren Gardner**[1,2,3], **Kimia Ghobadi**[1,2]*

**1** Department of Civil and Systems Engineering, Johns Hopkins University, Baltimore, Maryland, United States of America, **2** Center for Systems Science and Engineering, Johns Hopkins University, Baltimore, Maryland, United States of America, **3** Department of Epidemiology, Johns Hopkins Bloomberg School of Public Health, Baltimore, Maryland, United States of America

* kimia@jhu.edu

**Data Availability Statement:** All data used in this manuscript are sourced from publicly available datasets: 1. A County-level Dataset for Informing the United States' Response to COVID-19 (https://

## Abstract

The COVID-19 pandemic emphasized the need for pandemic preparedness strategies to mitigate its impacts, particularly in the United States, which experienced multiple waves with varying policies, population response, and vaccination effects. This study explores the relationships between county-level factors and COVID-19 mortality outcomes in the U.S. from 2020 to 2023, focusing on disparities in healthcare access, vaccination coverage, and socioeconomic characteristics. We conduct multi-variable rolling regression analyses to reveal associations between various factors and COVID-19 mortality outcomes, defined as Case Fatality Rate (CFR) and Overall Mortality to Hospitalization Rate (OMHR), at the U.S. county level. Each analysis examines the association between mortality outcomes and one of the three hierarchical levels of the Social Vulnerability Index (SVI), along with other factors such as access to hospital beds, vaccination coverage, and demographic characteristics. Our results reveal persistent and dynamic correlations between various factors and COVID-19 mortality measures. Access to hospital beds and higher vaccination coverage showed persistent protective effects, while higher Social Vulnerability Index was associated with worse outcomes persistently. Socioeconomic status and vulnerable household characteristics within the SVI consistently associated with elevated mortality. Poverty, lower education, unemployment, housing cost burden, single-parent households, and disability population showed significant associations with Case Fatality Rates during different stages of the pandemic. Vulnerable age groups demonstrated varying associations with mortality measures, with worse outcomes predominantly during the Original strain. Rural-Urban Continuum Code exhibited predominantly positive associations with CFR and OMHR, while it starts with a positive OMHR association during the Original strain. This study reveals longitudinal persistent and dynamic factors associated with two mortality rate measures throughout the pandemic, disproportionately affecting marginalized communities. The findings emphasize the urgency of implementing targeted policies and interventions to address

arxiv.org/abs/2004.00756) 2. CDC/ATSDR Social Vulnerability Index 2020. https://www.atsdr.cdc.gov/placeandhealth/svi/index.html. 3. An interactive web-based dashboard to track COVID-19 in real time: https://www.thelancet.com/journals/laninf/article/PIIS1473-3099(20)30120-1/fulltext 4. HHS. COVID-19 Reported Patient Impact and Hospital Capacity by Facility; 2020. Available from: https://healthdata.gov/Hospital/COVID-19-Reported-Patient-Impact-and-Hospital-Capa/anag-cw7u. 5. US COVID-19 Vaccination Tracking Georgetown University: http://www.vaccinetracking.us.

**Funding:** The authors received no specific funding for this work.

**Competing interests:** The authors have declared that no competing interests exist.

disparities in the fight against future pandemics and the pursuit of improved public health outcomes.

## Introduction

The World Health Organization declared COVID-19 a pandemic on March 11, 2020 [1, 2] which went on to pose an unprecedented public health challenge for the U.S. and countries worldwide. By the time the Biden administration announced the end of the COVID-19 public health emergency declarations on May 11, 2023 [3], multiple surges had resulted in a global case count exceeding 750 million and a death toll of approximately seven million worldwide, with more than 100 million cases and over a million deaths in the U.S. alone [4]. The dynamics influencing COVID-19 mortality rates have evolved during the three years of the pandemic due to different virus variants, shifting public health policies, new treatment options, vaccination roll-outs, and altering public responses. This study conducts a longitudinal analysis at the US county level to reveal the determinants of COVID-19 mortality outcomes, specifically Case Fatality Rate (CFR, ratio of COVID-19 attributed deaths to confirmed cases) and Overall Mortality to Hospitalization Rate (OMHR, ratio of total COVID-19 attributed deaths to confirmed hospitalizations), exploring how associations with these outcomes have changed or persisted throughout the pandemic.

While prior research has extensively examined COVID-19 death rates across different regions and demographic groups, focusing primarily on early pandemic stages and specific attributes such as socioeconomic factors [5–10], longitudinal studies capturing the evolving nature of the pandemic's impact remain less common, a few examples include [11–13]. Adding to the growing body of the literature, our study aims to fill the gap in understanding how determinants of COVID-19 mortality have dynamically changed over time, with a special focus on county-level factors in the United States from August 2020 to March 2023. Our approach builds upon the methods previously employed in COVID-19 mortality research, including machine learning models, linear regressions, and logistic analyses, by conducting a comprehensive county-level rolling horizon regression analysis. This methodological choice allows for a nuanced examination of temporal and spatial variations in mortality determinants, reflecting the complex interplay of factors influencing COVID-19 outcomes over the course of the pandemic. By doing so, we contribute insights into the persistence and evolution of these determinants, particularly highlighting disparities and healthcare access as critical factors.

A multitude of studies have undertaken investigations into a diverse range of attributes to the COVID-19 mortality measures focusing on demographic and/or geographical attributes, with particular emphasis on socioeconomic characteristics [9, 10, 14, 15], policy design during surges [16], and health care infrastructure and nursing home counts [16–19]. Among the potential COVID-19 mortality factors, disparities, whether rooted in healthcare infrastructure, socioeconomic factors, or demographic characteristics, have been observed to have a significant impact on the severity and outcomes of infectious diseases [20–22]. Disparities in mortality rates have been studied and distinct trends in severity, transmissibility, and vaccine response across various COVID-19 variants have been identified [23–26]. Existing work has examined contributing factors, such as health indicators [14, 16, 27], air pollution [14, 28–30], race [31], government effectiveness [18] and early vaccination effects [32, 33], mostly throughout the early stages of the pandemic. Another inseparable factor in studying pandemic outcomes includes healthcare capacity attributes, which are highly entangled with access to

healthcare and patient behaviors [34], subsequently attributed to healthcare outcomes. The literature has investigated the influential factors around healthcare capacity attributes such as hospitalization and ICU usage [35], early state analyses on the number of ICU beds [36], inpatient cases [37], and broader analysis on hospitalization trends [38] on COVID-19 outcomes.

A wide range of methods has been employed to study COVID-19 mortality including machine learning based models [6, 7], linear mixed models [16], linear regression models [8, 17, 18, 29, 30, 35], logistic regression [5, 31] and random effects models [27]. Recent studies on the temporal and/or spatial differences throughout the progression of the COVID-19 pandemic shed light on the inherent non-homogeneity of mortality and outcome patterns, including different outcomes from a social vulnerability lens [39] and shifts of higher mortalities from urban areas to more rural areas as the pandemic progresses [40]. The time intervals around vaccination campaigns have been recognized as pivotal in shaping these dynamics. Given the constantly evolving nature of pandemic severity over different time periods, exploring associations in a temporal manner is imperative.

The COVID-19 pandemic has exposed and exacerbated existing disparities, highlighting the urgent need to understand their role in shaping COVID-19 mortality rates [41, 42] while also providing a foundation for preparing for future unprecedented public health crises. While the existing COVID-19 outcomes literature predominantly focuses on the early stages of the pandemic, there is a need for longitudinal analyses of factors associated with COVID-19 outcomes, as these associations can exhibit dynamic shifts over time, partly in response to changes in human behavior, policy measures, and the evolving nature of the epidemic itself [43–49]. Understanding these shifts can firstly highlight factors that were persistently associate with COVID-19 mortality despite the changes in policy, treatments, vaccines, and behavior, possibly pointing to the need for fundamental solutions to address them in future pandemics, for instance, social vulnerability index has been shown to have positive association with COVID-19 mortality both in the literature [50–52] and our findings. Secondly, identifying factors that varied throughout the pandemic can inform decision-makers of potential trade-offs and impacts of varying policies and behavior. For instance, racial disparities are shown to have a positive association with COVID-19 mortality in the U.S. during the early part of the outbreak, with a declining trend during 2020 [43, 44]. Our results confirm this finding, with the additional observation that the association reverses direction and becomes consistently and significantly negative for a substantial portion of the later COVID-19 outbreak timeline after the initial waves. Similarly, the number of hospital beds per capita is deemed insignificant in its association with COVID-19 mortality in 2020 by Karmakar, Lantz, and Tipirneni (2021) [53]. While our results point to the same findings with respect to CFR for 2020, they show that this variable emerges as one of the most consistently significant explanatory factors for COVID-19 mortality in the later stages of the pandemic, highlighting the importance of hospital bed availability beyond the early parts of the pandemic and the initial lockdowns and reduced hospital operations. Our results also reveal other dynamically changing trends. The proportion of vulnerable age groups, including those younger than 17 and older than 65, showed associations with higher mortality measures before the prevalence of the Alpha variant and the start of vaccination roll-outs. However, these associations exhibited changing signs and were mostly insignificant for the remainder of the pandemic. The association between the population with a disability and mortality flipped from lower and mostly insignificant during the Original strain to higher mortality in the subsequent stages of the pandemic. The Rural-Urban Continuum Code demonstrated predominantly negative associations with Case Fatality Rate (CFR) and Overall Mortality to Hospitalization Rate (OMHR), while demonstrating a positive association with OMHR during the original strain. These examples underscore the importance of

conducting longitudinal analyses to capture the evolving associations between various factors and COVID-19 outcomes within the dynamic and ever-changing pandemic landscape.

This study specifically seeks to underscore how disparities, whether in healthcare infrastructure, socioeconomic conditions, or demographic profiles, have influenced COVID-19 mortality rates. Through our longitudinal analysis, we aim to identify persistent factors affecting mortality outcomes across different COVID-19 variants, e.g., the impact of social vulnerability and healthcare access, as well as identify factors whose influence has shifted, including group quarters population, population with disabilities, and Rural-Urban Continuum Codes (RUCC), possibly due to the effects of policy changes and public behavior adjustments.

In doing so, our work offers perspectives for public health responses, guiding resource allocation, and developing targeted interventions to mitigate the unequal burden of COVID-19 across communities. By addressing the dynamic shifts in the determinants of COVID-19 mortality and their associations over time, this study contributes to a more comprehensive understanding of the pandemic's impact, offering a foundation for better preparedness and response in the face of future public health crises.

## Materials and methods

In this study, we consider two COVID-19 mortality measures: Case Fatality Rate (CFR) and Overall Mortality to Hospitalization Rate (OMHR). CFR is a standard mortality measure defined as the ratio of COVID-19-attributed deaths to the number of confirmed cases (calculated as a percentage), and is a widely recognized measure of disease severity and prognosis, offering insights into the mortality risk associated with the virus within the broader population. To complement CFR and due to the lack of granular in-hospital death data at the county level, OMHR is introduced as a proxy measure and is defined as the ratio of total COVID-19-attributed deaths to the number of COVID-19 confirmed hospitalizations. While OHMR is different from the standard Hospital Fatality Rate, which only considers in-hospital deaths, it approximates the mortality risk among hospitalized cases to highlight care gaps and disease severity, as a significant portion of COVID-19 deaths occur in hospitals [54], though not exclusively.

These two measures complement each other to portray a more comprehensive picture of the progression of the pandemic since access to tests, reporting and data quality, vaccination coverage, and treatments changed during the span of the pandemic. These changes impacted both measures to a varying degree. CFR is affected by the quality and quantity of testing data, especially in the early stages of the pandemic when access to tests was limited and at the late stages of the pandemic when home tests became prevalent. OMHR, however, relies on the data reported by hospitals. While the quality of reported data is more consistent than CFR (as the data is reported by hospitals and collected by the United States Department of Health and Human Services (HHS) [55]), it is impacted by the availability of treatments and hospitals' definition of a COVID-19 patient.

We study the longitudinal association of these two mortality measures with a set of representative variables on different aspects of disparities and county-level factors during the COVID-19 pandemic from August 2020 to March 2023. Our research centers on examining the association between COVID-19 mortality measures and different aspects of disparities, and discerning any significant shifts as the dynamics around the pandemic changed. We utilize a rolling multi-variable linear regression analysis to capture variations in associations through the course of the pandemic, enabling us to investigate the progression and evolution of the observed relationships over time.

**Table 1. General categorization of the variables considered in the analyses.** The column Type indicates whether the variable is used as an explanatory or an outcome variable in the model. The columns Data and Level show the dataset's temporal and spatial granularity.

| Name | Description | Type | Data | Level |
|---|---|---|---|---|
| **Beds** | Number of staffed inpatient adult beds per 1000 people | expl. | time series | facility* |
| **SVI** | Social Vulnerability Indices (Overall or level I or II) | expl. | static | county |
| **65+ Percentage** | Population percentage of 65+ age group | expl. | static | county |
| **RUCC** | Rural Urban Continuum Code | expl. | static | county |
| **Vaccination Coverage** | Percentage of the fully vaccinated population | expl. | time series | county |
| **COVID-19 Attributes** | case fatality rate | outcome | time series | county |
| | Overall Mortality to Hospitalization Rate | outcome | time series | county |

*The data is aggregated to county level as outlined in Section Data Processing.

## Study design

We consider five factors that correlate with mortality rates as independent variables (as illustrated in Table 1). Our research hypothesis for each of the explanatory variables is the existence of an association (non-zero regression coefficient) between them and the outcome of COVID-19 mortality variables throughout the time frame of the study. We perform rolling linear regression analyses on a three-week long window, moving weekly, for each of the two mortality outcome variables (CFR and OMHR). Statistical significance of associations was assessed using the t-test, with a threshold of $\alpha = 0.05$, to reliably determine the presence of meaningful associations between variables and mortality outcomes within our linear regression framework.

To capture disparities, we consider three sets of explanatory variables, namely, social vulnerability indices, healthcare capacity, and vaccination coverage. We utilize the Centers for Disease Control and Prevention/Agency for Toxic Substances and Disease Registry (CDC/ATSDR) Social Vulnerability Indices (SVIs) [56] to assess community vulnerability. The Social Vulnerability Index (SVI) dataset is organized into three hierarchical levels of detail, as shown in Fig 1. The most aggregated level, Level I, consists of a single composite SVI score that summarizes the overall social vulnerability of a county. The next level, Level II, breaks down the SVI into four main themes: Socioeconomic Status, Household Characteristics, Racial & Ethnic Minority Status, and Housing Type & Transportation Each of these themes represents a distinct aspect of social vulnerability. The most granular level, Level III, further divides the four themes into a total of sixteen individual variables, providing more detailed information on specific factors contributing to each social vulnerability theme. These variables were selected to comprehensively address different dimensions of social and health disparities. To better understand how these different aspects of the SVI relate to COVID-19 mortality outcomes, we conduct three separate analyses, each corresponding to one of the three granularity levels described above. Together, these three levels of analyses allow for a more nuanced interpretation of the results, as the findings at each level complement and inform the others, providing insights into the complex relationships between various aspects of disparities and COVID-19 mortality outcomes.

As an indicator of healthcare access, we include county-level healthcare capacity measured by the number of staffed hospital beds per 1,000 people, referred to as beds for brevity. We estimate the total hospital bed capacity available to each county by factoring in a portion of the hospital beds within the county itself and those in neighboring counties, as outlined in the Data Sources and Processing Section. In addition to beds, we incorporate vaccination coverage

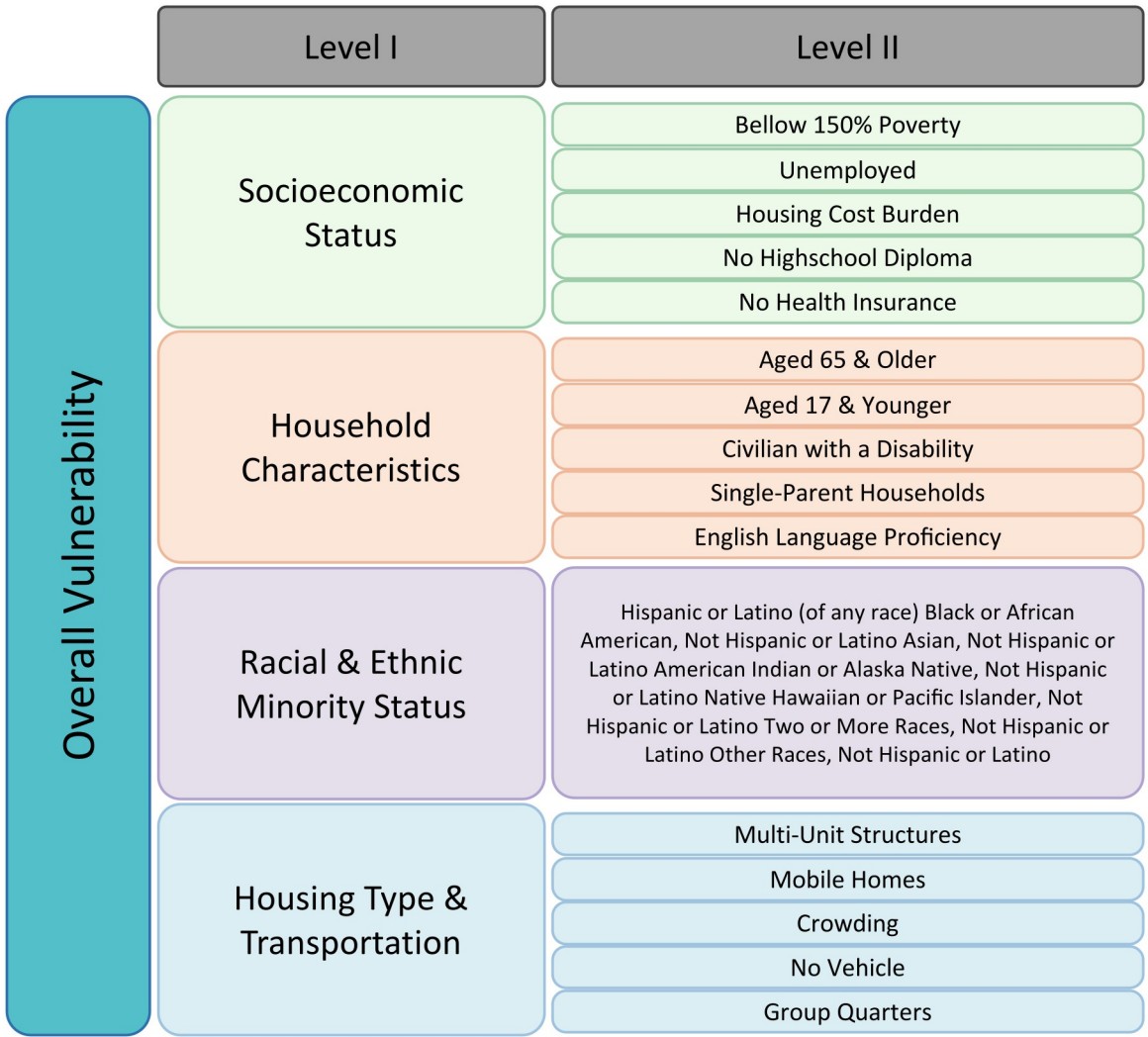

**Fig 1. Social Vulnerability Index composition: A hierarchical breakdown of socioeconomic, household, racial/ethnic, and housing/transportation themes of SVI.** CDC provides three levels of granularity for SVI. The overall SVI (Level I) is disaggregated into four themes for Level II and into 16 themes for Level III. In this study, we consider the same definitions of the vulnerability indices and perform three sets of analyses using each of the three levels (Figure modified from [56]).

by including the percentage of the population that has received COVID-19 vaccines. We include two additional factors to capture county's characteristics and population. We consider the population ratio of individuals aged 65 and above to account for the higher vulnerability of older adults to severe COVID-19 outcomes, as well as the rural-urban continuum code (RUCC) [57, 58] to account for any potential influence of rural-urban differences. Table 1 provides a summary of the variables, their role in our models, and the type and level of granularity of the dataset. More details on these variables are contained in the Data Processing section.

**Timeline.** While we intended to perform analyses for the entire duration of the pandemic, the period from August 2020 to March 2023 was selected based on data availability, to capture the most comprehensive view possible of the pandemic's evolution during significant phases of response and vaccination roll-out. The time series data on beds and hospitalizations which is collected by the United States Department of Health and Human Services (HHS) starts in

August 2020 [55]. Lockdowns and vastly different hospital operations make using static bed data from prior years inappropriate for the missing data in March to July 2020. Therefore, we limit the start of our analyses to August 2020 when the HHS data becomes available. We take steps to safeguard against the gradual growth of this data in 2020 as outlined in the Data Processing section. Similarly, the county-level data provided by the Johns Hopkins COVID Resource Center stopped collecting county-level cases and deaths on 10 March 2023 due to data unavailability and inconsistent reporting for several counties. Without a trusted county-level data source, we had to limit our analyses to March 2023 instead of May 2023.

## Methodological rigor: Ensuring data quality and accounting for variant influence

To maintain the integrity of our analyses and ensure the reliability of our findings, we have implemented robust data quality checks and adjusted our methodology to account for the influence of different COVID-19 variants.

**Data quality checks.** As the data reporting quality changed over time, we employ two outlier detection strategies and examine the variables for multicollinearity. To mitigate the influence of outliers, firstly, we restrict the analysis to include only those counties where the target variable's values fall below the 95th percentile. Secondly, we calculate Cook's distance for each analysis, excluding counties with values exceeding three times the mean Cook's distance. This approach effectively reduces distortion from extreme outliers, affecting less than 9% of counties in each analysis. The detailed time series representation of excluded counties is available in S1 Fig in the Supporting Information Section.

To check multicollinearity, we examine pairwise Pearson correlations and Variance Inflation Factors (VIFs). For analyses incorporating the first two Social Vulnerability Index (SVI) levels, Pearson correlation coefficients do not exceed 0.6, and for the level III analysis, they remain below 0.7. Across all analyses, VIF scores are under 5, indicating minimal multicollinearity, as illustrated in S2–S4 Figs in the Supporting Information section. Despite the 65 + Age variable being part of the level III SVI variables, it shows low correlations with overall or level II SVI variables (PCC < 0.3), and hence is retained in the first two levels of analysis due to its significance in explaining mortality outcomes but excluded from level III.

**Variant detection.** Given the distinct mortality rates attributed to different COVID-19 variants, we segmented our analysis timeline into four variant-specific intervals: Original strain or Original for brevity (August 2020 to April 2021), Alpha (April to June 2021), Delta (June to December 2021), and Omicron (December 2021 to March 2023). These intervals are defined based on when each variant represented over half of the reported COVID-19 cases in the United States on state level [59], and facilitate better interpretability of the results. To ascertain prevalence, we calculated a population-weighted sum of the sequences corresponding to each variant. Each day, the variant with the highest weighted sum of sequences was deemed the prevailing variant within that time frame.

**Time lag considerations.** It is critical to account for the time lags between COVID-19 contraction, hospitalization, and death. Similar to the findings, we implement a one-week lag between infection to hospitalization and one week between hospitalization to death in our analysis [60–64]. Therefore, admissions time series data is lagged by one week relative to deaths, with cases and vaccination time series data lagged by two weeks, as demonstrated in Fig 2. To reduce noise in the time series data, we apply three-week moving averages. We then conduct weekly regression analyses for August 2020 to March 2023 and obtain a time series of coefficients and significance levels for each pair of our outcome and explanatory variables, that showcase the progression of their association through the study time horizon.

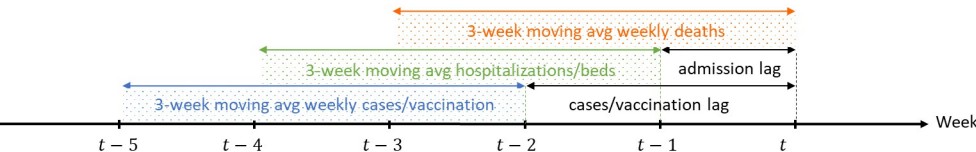

**Fig 2. Schematics of the considered time lags and the moving averages (patterned windows) that are used at each snapshot of our analyses.** Parameters used at timepoint $t$ are estimated based on the prior weeks' cases and fatalities.

## Data processing

This section outlines the data processing steps taken to prepare the datasets for analysis, highlighting their origins, treatment, and significance for our study. The datasets span various sources, reflecting the multifaceted nature of COVID-19's impact across the United States. These include static and dynamic (time series) variables, each necessitating specific handling to ensure accuracy and relevance to our analyses. Table 1 provides an overview of all variables considered in the study. Table 2 summarizes the distributions of the static explanatory variables, and Fig 3 demonstrates the distribution of the time-dependent explanatory and outcome variables as box plots.

### Data sources and handling

**Static variables**. The Rural-Urban Continuum Code (RUCC) and age datasets are sourced from [58]. The SVI variables are sourced from the CDC [56], which considers three nested levels of granularity. Level I contains a single overall index. Level II is categorized into four primary themes: socioeconomic status, household characteristics, minority status, and housing type and transportation. Level III breaks down these themes further into sixteen variables as illustrated in Fig 1. In our analysis, we examine the single score of Level I, the four main theme indices of Level II, and the sixteen variables of Level III in three distinct models, and investigate the unique contributions of each component.

**Dynamic variables**. *COVID-19 attributes*. We sourced time series data on cases and mortality on a county level from the COVID-19 Dashboard provided by the Johns Hopkins Center for Systems Science and Engineering (CSSE) and COVID Resource Center [65] and aggregated the time series data on a weekly basis. We note that this data is only available until March 2023, which limits our analyses to this date.

*Healthcare capacity*. We consider the total number of beds as a measure of healthcare access in each county and source this variable and confirmed COVID-19 admissions from a dataset published by the United States Department of Health and Human Services (HHS) [55]. The dataset provides weekly time series reports of hospital attributes at the facility level from July 2020 onwards. It includes data on 4,669 short-term and critical access hospitals across the nation.

**Table 2. Distribution of static variables selected for this study.** We only showcase the overall SVI since all SVI themes have uniform distributions.

|  | Mean | Std | Min | 25% | 50% | 75% | Max |
|---|---|---|---|---|---|---|---|
| **65+ Percentage** | 19.31 | 4.62 | 4.83 | 16.35 | 18.96 | 21.75 | 57.59 |
| **RUCC** | 4.95 | 2.68 | 1.00 | 2.00 | 6.00 | 7.00 | 9.00 |
| **SVI** | 0.50 | 0.29 | 0.00 | 0.26 | 0.51 | 0.75 | 1.00 |

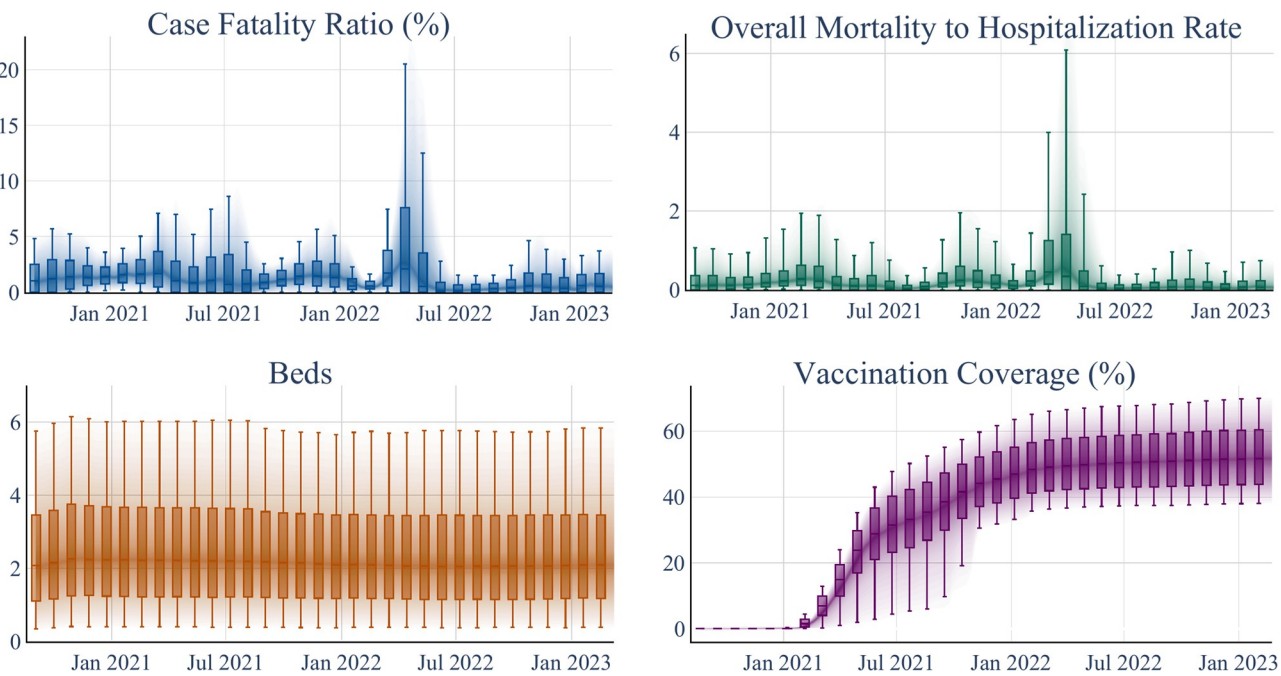

**Fig 3. Temporal trends in COVID-19 case fatality rate, overall mortality to hospitalization rate, beds and vaccination coverage.** Box plots represent 10, 25, 50, 75, and 90 percentiles of the distribution of each variable, aggregated in 4-week windows.

*Vaccination coverage.* We analyze the time series data representing the population's ratio of fully vaccinated individuals. The vaccination rate data utilized in our study is derived from Georgetown University's U.S. COVID-19 Vaccination Tracking website [66]. This dataset offers enhanced accuracy by amalgamating data from both the CDC and state reporting sources. In our study, the term "complete vaccination rate" signifies the cumulative proportion of the overall population that has received either two doses of the Pfizer or Moderna COVID-19 vaccines or one dose of the J&J COVID-19 vaccine. Vaccination coverage distributions are shown in Fig 3.

## Data cleaning and preparation

The COVID-19 attributes data is processed to provide weekly time series and any negative values is replaced with interpolated values. Fig 3 shows the time series distribution of case fatality and overall mortality to hospitalization rates. To determine the initial inclusion of counties in our analysis, we considered their initial three recorded COVID-19 deaths as the threshold for inclusion in the study sample. This criterion ensured a standardized starting point for each county's data inclusion in our analyses.

**Missing data estimation**. Approximately 75% of hospitals had begun consistently reporting their COVID-19 patient and capacity to the HHS on a weekly basis by August 2020. However, roughly 25% of bed data is missing in early August 2020, with the percent of missing values gradually decreasing to less than 2% by January 2021. Similarly, about 20% of the admission data is missing at the beginning, decreasing to below 3% by January 2021. The initial missing data in the early weeks of reporting may be attributed to hospitals needing time to establish and streamline the reporting processes required by HHS. To estimate some of the missing data, for each facility with missing reports, we use the available entries of the same date from other facilities in the same county, if any, using formulation 1. Following this estimation,

missing bed entries are decreased to under 15% initially, and below 1% after January 2021, and missing admissions are reduced to less than 12% initially, and below 1% for most parts after January 2021.

$$\text{missing value} = \frac{\text{point value of county facilities} \times \text{mean of missing entry facility}}{\text{mean of available data}} \quad (1)$$

To estimate hospital capacity and COVID-19 hospitalizations at the county level, we determine the extent to which each hospital serves each county. We employ the Hospital Service Area File (HSAF) published by the Centers for Medicare and Medicaid Services [67]. This data provides the total number of Medicare patients at each hospital, categorized by the zip code of their residence. We utilize the 2020 version of the HSAF, since it contains more available information compared to more recent versions. Initially, we aggregated HSAF data from zip code to county level. Subsequently, we assessed hospital service in each county by determining the proportion of hospital patients originating from that county. We estimated county hospital capacity by weighting each hospital's capacity based on its county-specific patient proportion. Employing a parallel methodology, we also projected COVID-19 hospitalizations for each county.

**Outlier detection**. The Hampel outlier detection method is applied to the hospital beds and admissions time series data using a sliding window of 8 weeks and a threshold of 6 median absolute deviations. The dynamic distribution of the number of beds remains relatively fixed after these modifications (refer to Fig 3).

## Results and discussion

In this section, we detail the results of the three levels of analysis. In reporting the results, we highlight how the relationships between various explanatory variables and COVID-19 mortality outcomes evolved throughout the pandemic. We focus on the dynamic associations over time, supported by regression analyses. We first present the results for the overall SVI. We then delve into the four themes of SVI, at both level II and level III granularity (see S5–S8 Figs, and S1 and S2 Tables, in supporting information section for level III results). The time series of R-squared values for each of the three levels of analyses are also illustrated in S9 Fig and S3 Table in the Supporting Information Section.

### Hospital bed availability

The analysis reveals an evolving relationship between hospital bed availability and COVID-19 mortality outcomes. Initially, during the Original variant phase, the number of hospital beds did not significantly influence the Case Fatality Rate (CFR). However, starting with the Alpha variant, a consistent negative association emerged, indicating that increased hospital capacity significantly correlates with reduced mortality rates. This trend persisted through the Delta and Omicron variants, with p-values ranging between <0.001 and 0.04, as illustrated in Fig 4. In this plot, the coefficient of the explanatory variable in relation to the targeted outcome variable is shown on the y-axis, with the coefficient curves smoothed using a three-week moving average. The 95[th] percentile of the regression coefficient is shaded around each curve. Positive significant coefficients ($P < 0.05$), above the zero line, are shaded in blue, while negative significant coefficients below the zero line are represented in orange. Additionally, each dominant variant interval is distinctly marked with solid vertical lines, segmenting the time horizon into four distinct intervals: Original, Alpha, Delta, and Omicron variants. Other plots in this study follow the same structure. The Overall Mortality to Hospitalization Rate (OHMR) reveals

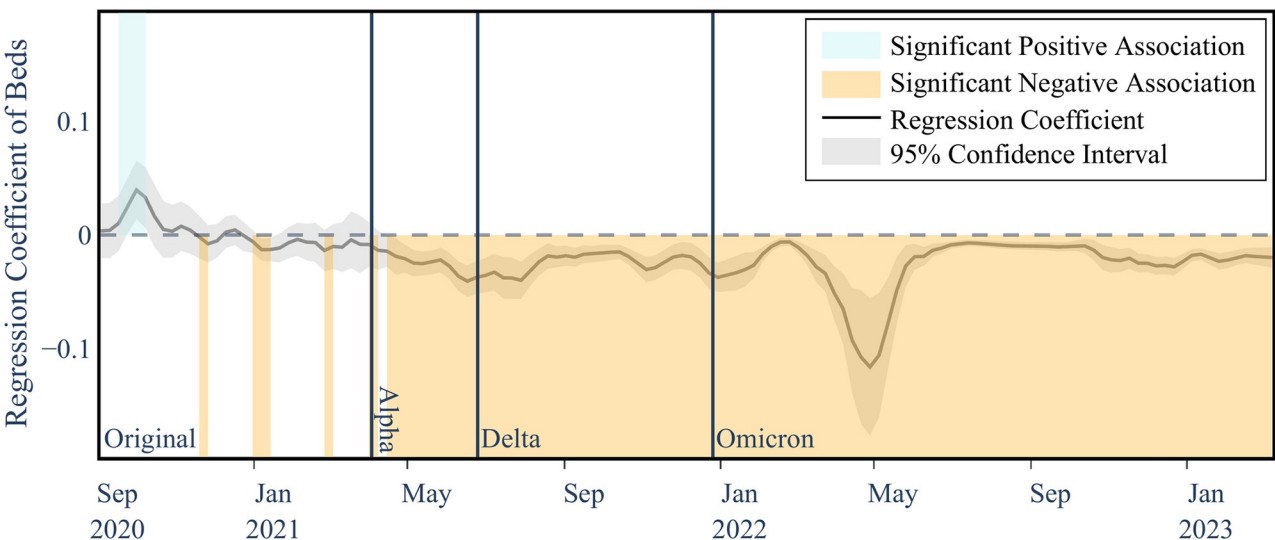

**Fig 4. Example of temporal association visualization between number of hospital beds and COVID-19 case fatality rate.** Temporal associations between the number of beds (per a 1000) and case fatality rate as an example of how the results are represented in this paper. The plotted line represents the regression coefficient values between beds and case fatality rate. The gray shaded area represents the 95% confidence interval for the coefficient curve at each point in time. For clearer representation, any time interval in which a positive or negative significant association is observed (i.e., above or below the zero line, respectively) is colored blue or orange, respectively. Each dominant variant interval is separated using solid vertical lines, dividing the time horizon into four intervals of Original, Alpha, Delta, and Omicron variants. The curves are smoothed using a three-week moving average to reduce extraneous noise and highlight the underlying trend of the associations. All subsequent figures follow the same representation format.

similar trends (P<0.001) and beds keeping their significant negative association even during the early stages of the pandemic (first row in Fig 5).

The results emphasize the importance of robust healthcare infrastructure in managing severe cases. Counties with higher hospital bed availability were better equipped to manage severe cases, resulting in lower case fatality rates and lower overall mortality to hospitalization rates. For more details on the CFR and OMHR regression results refer to Tables 3 and 4.

## Vaccination coverage

Post-vaccine rollout, regions with higher vaccination rate typically experienced lower CFR and OMHR, particularly notable during the Delta and early Omicron phases (see Fig 5 and Tables 3 and 4). However, this relationship showed variability, with strongest association during Delta and the first half of the Omicron prevalence periods but weakened during Alpha and the second half of Omicron, likely influenced by the emergence of new variants and changing public health policies. Notably, the relationship briefly reversed following the Delta surge (lasting for only about 23% and 27% of the weeks for CFR and OMHR, respectively). The observed predominantly negative correlation between vaccination rates and COVID-19 mortality aligns with existing research, strengthening the notion that vaccinations were effective in reducing COVID-19 mortality [68]. The occasional shifts in associations can be attributed to the rise of new variants, notably the Delta and Omicron variants, as our vaccination dataset includes the initial rounds of vaccinations and does not account for the booster doses. Moreover, the evolving landscape of public health policies likely played a role in shaping mortality trends, especially as the pandemic evolved. Despite these factors suggesting a complex interplay between vaccination, variant susceptibility, and healthcare strategies in determining COVID-19

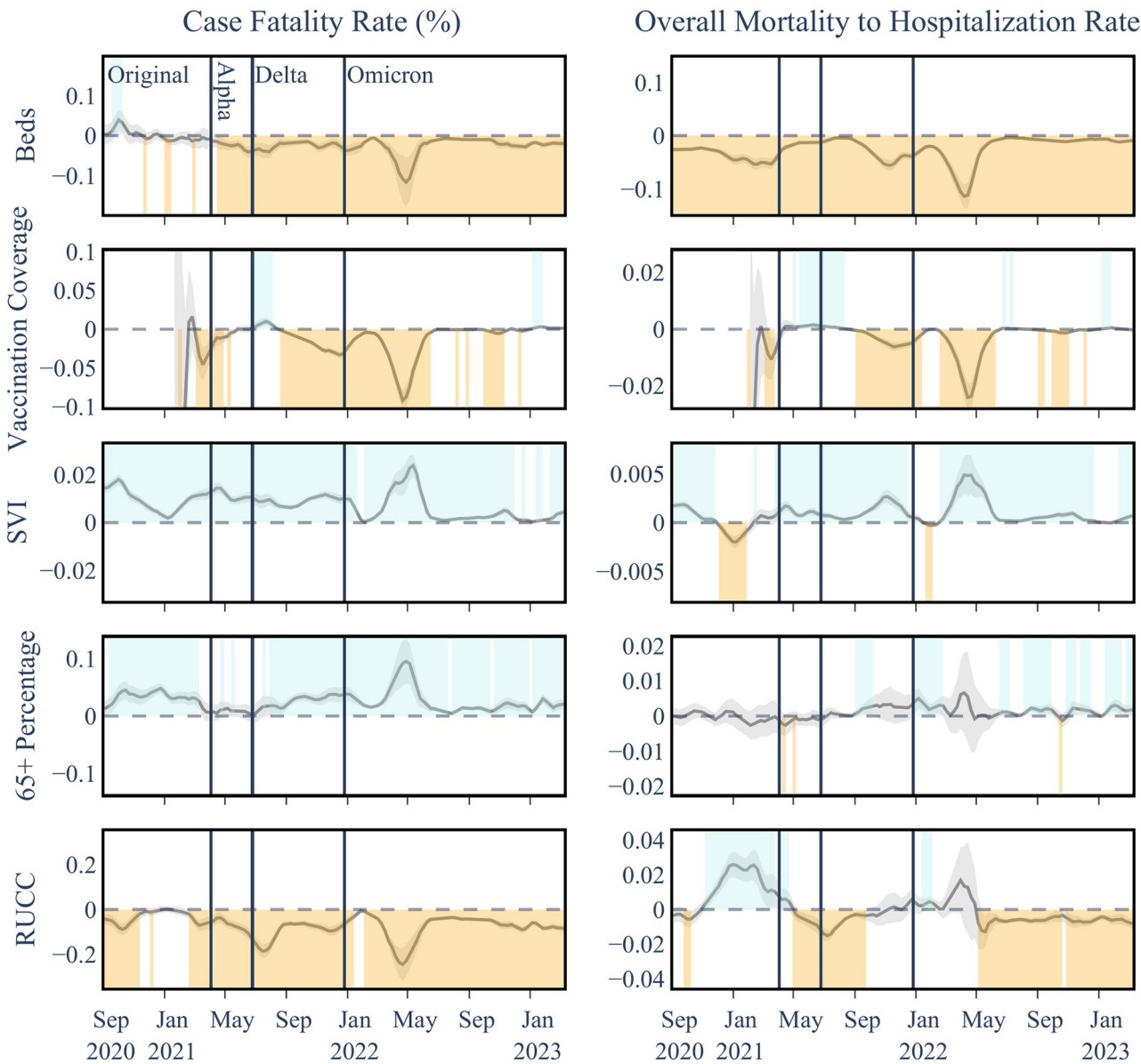

**Fig 5. Temporal associations between COVID-19 outcome and explanatory variables in Level I analysis.** Shaded areas above and below the horizontal zero line indicate significant positive and negative associations for each date, respectively, within a 95 percent confidence interval.

mortality rates, the predominantly negative associations observed between vaccination coverage and mortality outcomes highlight the protective effect of vaccinations against severe COVID-19 outcomes, even in the absence of a more comprehensive vaccination coverage dataset.

## Age group 65+ percentage

The percentage of the population aged 65 and over consistently showed a strong positive correlation with CFR (Table 3), although briefly insignificant before widespread vaccine distribution. This period coincides with the early stages of vaccination campaigns, which prioritized

**Table 3. Summary of key results for CFR with SVI Level I.** For each variable and variant period, the overall association, the week % of statistical significance, the coefficients, and the p-values are provided. Those with statistical significance during more than half of the weeks are highlighted in bold.

| Variable | Variant | Assoc. | Weeks significant | | | Coeffs. avg (std) | | P-value* avg (std) | |
|---|---|---|---|---|---|---|---|---|---|
| | | | - | + | Total | - | + | - | + |
| Beds | Original | Insig. | 13% | 10% | 31 | -0.02 (0.01) | 0.04 (0.01) | 0.02 (0.02) | 0.02 (0.03) |
| | Alpha | **Neg.** | **92%** | 0% | 12 | -0.03 (0.01) | | 2e-03 (5e-03) | |
| | Delta | **Neg.** | **100%** | 0% | 26 | -0.02 (0.01) | | 9e-04 (2e-03) | |
| | Omicron | **Neg.** | **100%** | 0% | 63 | -0.03 (0.03) | | 2e-03 (7e-03) | |
| Vaccination Coverage | Original | Insig. | 45% | 0% | 11 | -0.09 (0.1) | | 5e-04 (1e-03) | |
| | Alpha | Insig. | 42% | 0% | 12 | -0.01 (0.01) | | 0.02 (0.02) | |
| | Delta | **Neg.** | **69%** | 23% | 26 | -0.02 (0.01) | 0.01 (3e-3) | 3e-04 (1e-03) | 0.02 (0.02) |
| | Omicron | **Neg.** | **54%** | 5% | 63 | -0.02 (0.03) | 3e-3 (7e-4) | 5e-03 (1e-02) | 3e-03 (3e-03) |
| SVI | Original | **Pos.** | 0% | **100%** | 31 | | 0.01 (5e-3) | | 1e-04 (5e-04) |
| | Alpha | **Pos.** | 0% | **100%** | 12 | | 0.01 (2e-3) | | 7e-19 (3e-18) |
| | Delta | **Pos.** | 0% | **100%** | 26 | | 0.01 (2e-3) | | 2e-09 (1e-08) |
| | Omicron | **Pos.** | 0% | **86%** | 63 | | 0.01 (0.01) | | 3e-03 (1e-02) |
| 65+ Percentage | Original | **Pos.** | 0% | **87%** | 31 | | 0.03 (0.01) | | 2e-03 (9e-03) |
| | Alpha | Insig. | 0% | 17% | 12 | | 0.02 (0.01) | | 0.01 (0.02) |
| | Delta | **Pos.** | 0% | **85%** | 26 | | 0.03 (0.01) | | 3e-03 (1e-02) |
| | Omicron | **Pos.** | 0% | **95%** | 63 | | 0.03 (0.02) | | 1e-03 (5e-03) |
| RUCC | Original | **Neg.** | **58%** | 0% | 31 | -0.06 (0.02) | | 6e-03 (1e-02) | |
| | Alpha | **Neg.** | **100%** | 0% | 12 | -0.07 (0.03) | | 7e-03 (2e-02) | |
| | Delta | **Neg.** | **100%** | 0% | 26 | -0.1 (0.04) | | 3e-11 (1e-10) | |
| | Omicron | **Neg.** | **95%** | 0% | 63 | -0.08 (0.06) | | 2e-06 (1e-05) | |

* All provided p-values are statistically significant for $\alpha$ = 0.05.

more vulnerable groups, particularly those aged 65 and older. It is plausible that the initial higher vaccination rates in this age group contributed to the temporary decrease in significance observed during the Alpha variant. This suggests that the focused vaccination efforts might have had a protective effect on the older population, temporarily altering the established correlation between age and COVID-19 fatality rates. The association with OMHR was less significant and less consistent with only Omicron showing a positive association (Table 4), potentially hinting that while 65+ endured a higher mortality rate when normalized to the number of COVID-19 cases and less so when normalized to the number of hospitalized COVID-19 patients, which likely includes many older patients within itself.

## Rural-urban continuum code

This factor exhibited a mostly negative correlation with CFR and a higher mortality in urban counties (Fig 5), with some variation during the Original and Omicron variants. For OMHR, RUCC showed a more complex shifting pattern between positive and negative. Our analysis reveals that generally, urban counties faced higher mortality rates, with an interesting deviation during the initial phase of the Original variant: a positive correlation emerged between RUCC and OMHR. This could be indicative of the unique challenges rural areas encountered in the early response to the pandemic, as also noted in the findings documented by the CDC Museum COVID-19 Timeline as of August 4, 2020 [1]. The shifting trends in the relationship between RUCC and COVID-19 mortality outcomes, as illustrated in Fig 6, highlight the dynamic impact of the pandemic across different types of communities.

**Table 4. Summary of key results for OMHR with SVI Level I.** For each variable and variant period, the overall association, the week % of statistical significance, the coefficients, and the p-values are provided. Those with statistical significance during more than half of the weeks are highlighted in bold.

| Variable | Variant | Assoc. | Weeks significant | | | Coeffs. avg (std) | | P-value* avg (std) | |
|---|---|---|---|---|---|---|---|---|---|
| | | | - | + | Total | - | + | - | + |
| Beds | Original | **Neg.** | **100%** | 0% | 31 | -0.0367 (0.0117) | | 5e-23 (2e-22) | |
| | Alpha | **Neg.** | **100%** | 0% | 12 | -0.0155 (0.0041) | | 8e-17 (2e-16) | |
| | Delta | **Neg.** | **100%** | 0% | 26 | -0.0260 (0.0194) | | 9e-12 (4e-11) | |
| | Omicron | **Neg.** | **100%** | 0% | 63 | -0.0224 (0.0289) | | 2e-11 (2e-10) | |
| Vaccination Coverage | Original | Insig. | 45% | 0% | 11 | -0.5722 (1.2141) | | 1e-02 (1e-02) | |
| | Alpha | **Pos.** | 0% | **58%** | 12 | | 0.0012 (0.0003) | | 7e-03 (8e-03) |
| | Delta | **Neg.** | **62%** | 27% | 26 | -0.0040 (0.0021) | 0.0007 (0.0002) | 2e-04 (9e-04) | 2e-03 (4e-03) |
| | Omicron | Insig. | 43% | 8% | 63 | -0.0067 (0.0079) | 0.0004 (0.0001) | 2e-03 (5e-03) | 0.01 (0.02) |
| SVI | Original | Mixed | 26% | 48% | 31 | -0.0014 (0.0006) | 0.0011 (0.0006) | 6e-04 (1e-03) | 9e-03 (1e-02) |
| | Alpha | **Pos.** | 0% | **100%** | 12 | | 0.0011 (0.0004) | | 1e-05 (3e-05) |
| | Delta | **Pos.** | 0% | **96%** | 26 | | 0.0012 (0.0008) | | 4e-04 (2e-03) |
| | Omicron | **Pos.** | 3% | **78%** | 63 | -0.0003 (0.0001) | 0.0013 (0.0015) | 0.01 (0.01) | 4e-03 (9e-03) |
| 65+ Percentage | Original | Insig. | 0% | 0% | 31 | | | | |
| | Alpha | Insig. | 17% | 0% | 12 | -0.0027 (0.0012) | | 0.04 (0.02) | |
| | Delta | Insig. | 0% | 19% | 26 | | 0.0023 (0.0006) | | 0.02 (0.02) |
| | Omicron | **Pos.** | 2% | **52%** | 63 | -0.0021 (nan) | 0.0024 (0.0011) | 0.03 | 0.01 (0.01) |
| RUCC | Original | **Pos.** | 6% | **65%** | 31 | -0.0064 (0.0006) | 0.0179 (0.0074) | 8e-03 (7e-03) | 5e-03 (1e-02) |
| | Alpha | **Neg.** | **67%** | 17% | 12 | -0.0067 (0.0025) | 0.0069 (0.0002) | 7e-03 (2e-02) | 0.02 (0.01) |
| | Delta | **Neg.** | **50%** | 0% | 26 | -0.0076 (0.0050) | | 2e-03 (5e-03) | |
| | Omicron | **Neg.** | **68%** | 5% | 63 | -0.0065 (0.0024) | 0.0045 (0.0004) | 4e-03 (1e-02) | 3e-03 (1e-03) |

\* All provided p-values are statistically significant for $\alpha$ = 0.05.

## Social Vulnerability Index

Regions with higher scores on the Social Vulnerability Index (SVI) exhibited worse COVID-19 outcomes across all variants, highlighting increased susceptibility in socially vulnerable communities. SVI showed a similar persistent positive relationship with OMHR, which was most pronounced during Alpha and Delta. The significant and persistent positive correlation between SVI and COVID-19 mortality outcomes highlights the increased susceptibility of socially vulnerable communities to fatalities associated with COVID-19. This susceptibility can be attributed to several factors prevalent in these communities, e.g., a higher occurrence of underlying health conditions or a limited access to healthcare, essential resources, and crucial information for effective COVID-19 mitigation. These combined factors not only intensify the rate of infection but also contribute to a higher burden of mortality in these communities, underlining the critical need for targeted interventions and support in areas with elevated social vulnerability. These associations are further studied in our subsequent analyses as outlied in the following Section.

## Level II and Level III analyses

In the second set of our analysis on **SVI Level II**, which includes four aggregate SVI theme indices (see Fig 1), we observed varying associations across different socioeconomic, demographic, and environmental factors. Socioeconomic Status (SVI 1) and Household Characteristics (SVI 2) predominantly demonstrated positive associations with both COVID-19 Case Fatality Rate (CFR) and Overall Mortality to Hospitalization Ratio (OMHR) throughout the

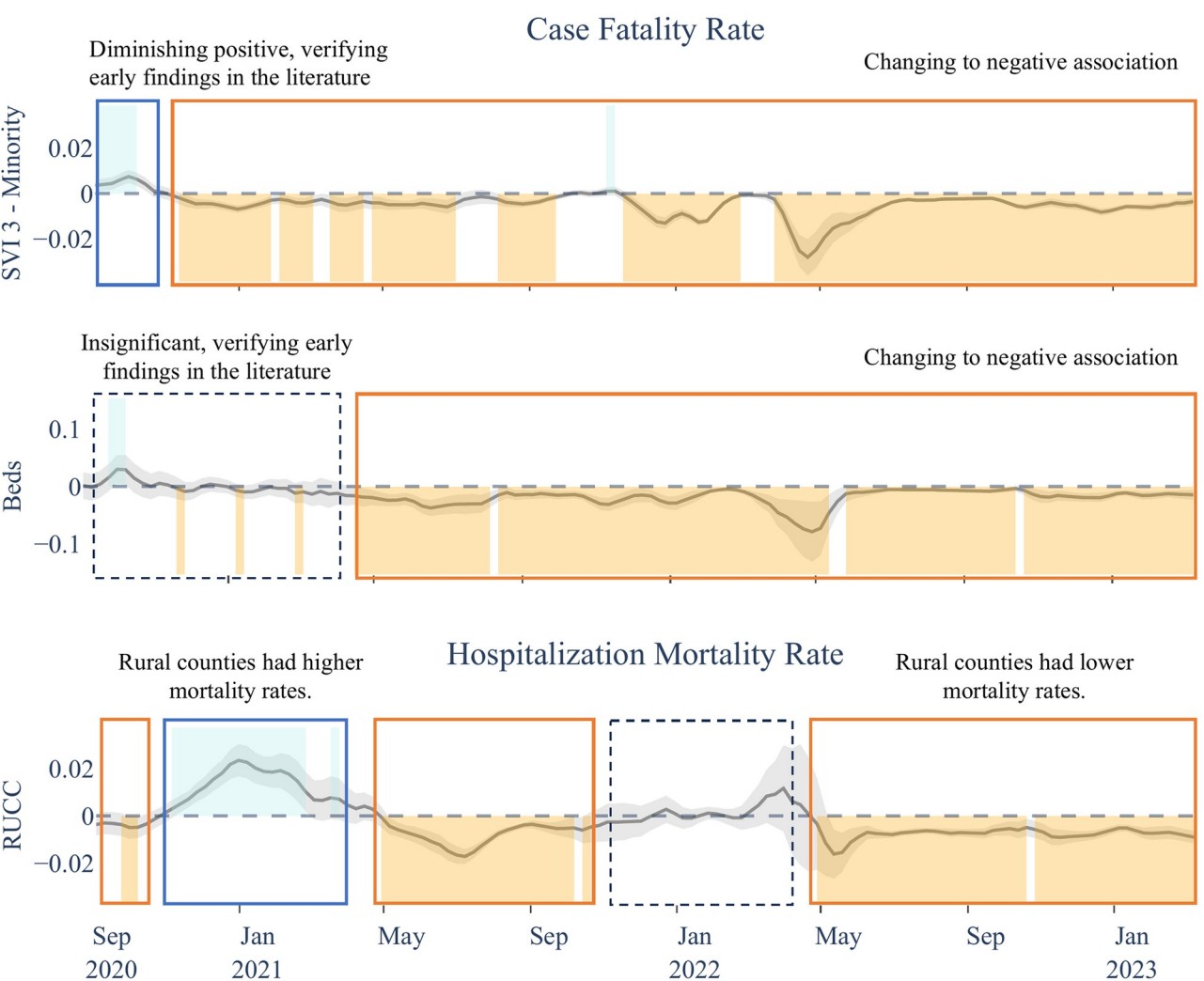

**Fig 6. Time-varying nature of COVID-19 mortality associations for select explanatory variables.** Examples of the time-varying nature of the associations—**SVI 3**: Literature has found evidence on a diminishing positive association between minority population and COVID-19 mortality rates in 2020 [43, 44]. Our results verify this observation, showing a positive association between SVI theme 3 and CFR in 2020 that diminishes into a negative association and stays predominantly negative through the rest of the pandemic. **Beds**: Early on literature finds beds per capita as an insignificant explanatory variable to COVID-19 mortality [53], which is captured in our results and is extended to a significant and persistent negative association starting May 2022. **RUCC**: CDC Museum COVID-19 Timeline on August 4th of 2020, highlights the challenges people living in rural areas have facing COVID-19 which can lead to higher modality rates. Our results showcase changing trends in the association between RUCC and OMHR, highlighting a positive association through the end of 2020 and early 2021, while showing insignificance or negative associations in other periods.

pandemic. In contrast, the Racial and Ethnic Minority Status factor (SVI 3) showed an initial positive association with mortality outcomes, which shifted to predominantly negative correlations from late 2020 onwards. The Housing Type and Transportation factor (SVI 4) exhibited a largely positive association with CFR but less consistent relationships with OMHR. Delving deeper into the individual **Level III** variables comprising these SVI themes provided important insights into the specific socioeconomic, demographic, and environmental factors potentially driving disproportionate COVID-19 impacts on vulnerable communities as the pandemic evolved. The results of these granular analyses on SVI are presented in this section.

**Socioeconomic status (SVI theme 1).**   Socioeconomic Status showed persistent positive associations with higher CFR throughout the pandemic and with OMHR following the

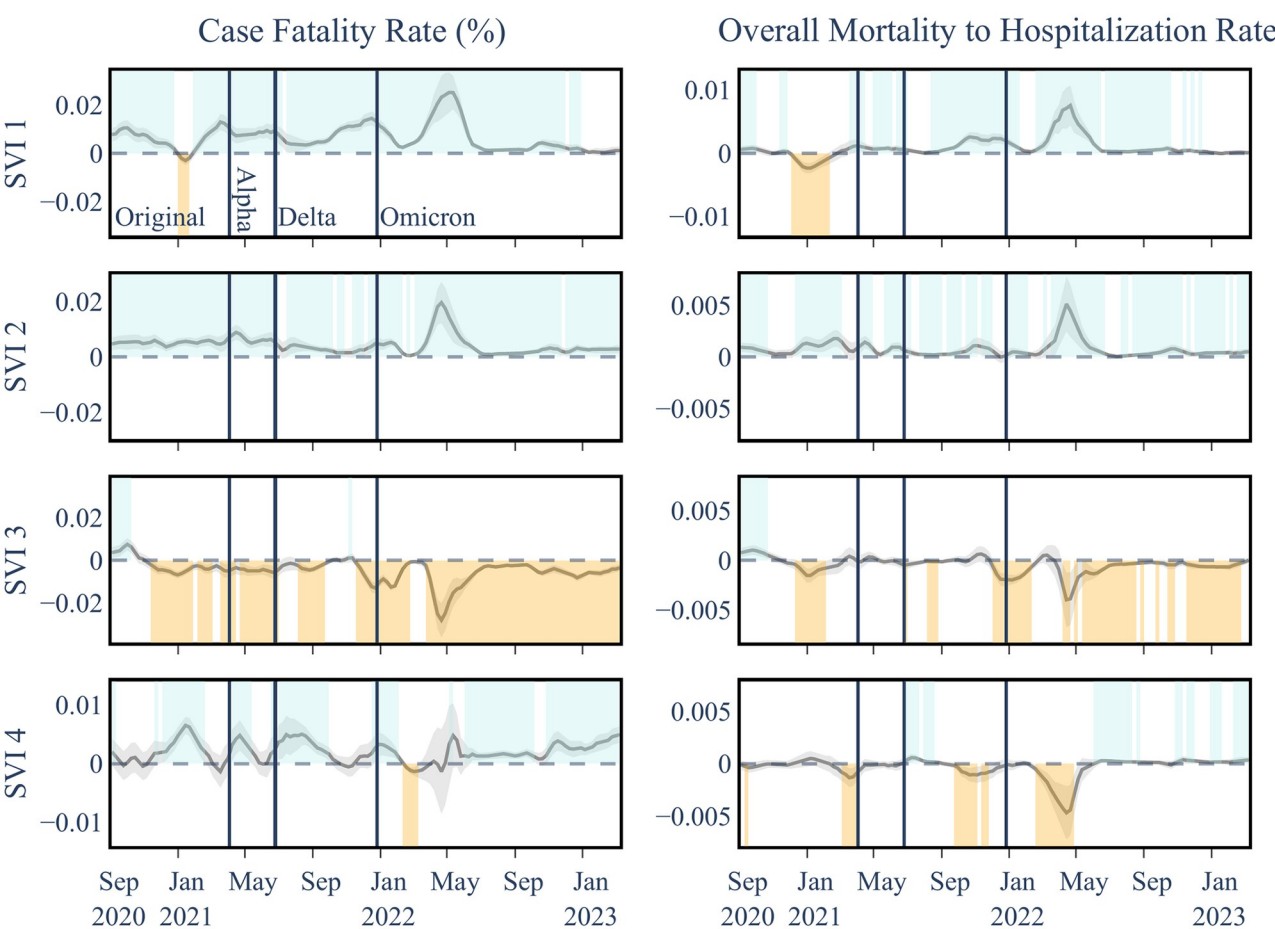

**Fig 7. Temporal associations between COVID-19 outcomes and SVI theme variables in Level II analysis.** Considering each SVI theme including **SVI 1: Socioeconomic Status**, **SVI 2: Household Characteristics**, **SVI 3: Racial & Ethnic Minority Status**, and **SVI 4: Housing Type & Transportation**. SVI 1 and 2 show almost persistent positive associations with both COVID mortality outcomes while SVI 3 and 4 show a mix of positive and negative associations with COVID mortality outcomes.

emergence of the Alpha variant (see Fig 7 and Tables 5 and 6). This finding suggests that counties with lower socioeconomic status experienced disproportionately worse COVID-19 mortality outcomes, particularly as the pandemic progressed. A closer examination of the Level III variables within this theme (see S5 Fig and S1 Table) revealed key drivers including Below 150% Poverty and No High School Diploma during the original strain and Alpha periods, transitioning to Unemployment and Housing Cost Burden during later stages. These shifts highlight how the economic fallout of the pandemic, including job losses and housing instability, played an increasingly significant role in shaping COVID-19 mortality disparities. Lack of Health Insurance also displayed positive associations with CFR during the original strain, Delta and early Omicron periods (see S5 Fig), demonstrating a fairly consistent positive association between lack of health insurance and higher CFRs, and underscoring how robust insurance coverage could play a role in mitigating the severe impacts of a pandemic.

**Household characteristics (SVI theme 2).** Household Characteristics maintained a consistent positive association with both CFR and OMHR throughout the pandemic (Fig 7 and Tables 5 and 6). An examination of the Level III variables within this theme revealed that vulnerable age groups, specfically Age 65 and Older and Age 17 and Younger, showed heightened

**Table 5. Summary of key results for CFR with SVI Level II.** For each variable and variant period, the overall association, the week % of statistical significance, the coefficients, and the p-values are provided. Those with statistical significance during more than half of the weeks are highlighted in bold.

| Variable | Variant | Assoc. | Weeks significant | | | Coeffs. avg (std) | | P-value* avg (std) | |
|---|---|---|---|---|---|---|---|---|---|
| | | | - | + | Total | - | + | - | + |
| SVI 1—Socioeconomic Status | Original | **Pos.** | 10% | **84%** | 31 | -0.0030 (0.0006) | 0.0075 (0.0033) | 2e-03 (2e-03) | 1e-03 (4e-03) |
| | Alpha | **Pos.** | 0% | **100%** | 12 | | 0.0082 (0.0013) | | 2e-05 (5e-05) |
| | Delta | **Pos.** | 0% | **96%** | 26 | | 0.0079 (0.0041) | | 4e-03 (1e-02) |
| | Omicron | **Pos.** | 0% | **83%** | 63 | | 0.0068 (0.0075) | | 2e-03 (7e-03) |
| SVI 2—Household Characteristics | Original | **Pos.** | 0% | **100%** | 31 | | 0.0051 (0.0009) | | 1e-03 (5e-03) |
| | Alpha | **Pos.** | 0% | **100%** | 12 | | 0.0066 (0.0015) | | 7e-06 (2e-05) |
| | Delta | **Pos.** | 0% | **77%** | 26 | | 0.0031 (0.0011) | | 8e-03 (1e-02) |
| | Omicron | **Pos.** | 0% | **95%** | 63 | | 0.0041 (0.0046) | | 3e-03 (8e-03) |
| SVI 3—Racial & Ethnic Minority Status | Original | **Neg.** | **55%** | 19% | 31 | -0.0049 (0.0013) | 0.0057 (0.0023) | 4e-04 (1e-03) | 8e-03 (2e-02) |
| | Alpha | **Neg.** | **92%** | 0% | 12 | -0.0049 (0.0011) | | 5e-03 (1e-02) | |
| | Delta | **Neg.** | **50%** | 4% | 26 | -0.0060 (0.0040) | 0.0020 (nan) | 2e-03 (5e-03) | 0.04 (nan) |
| | Omicron | **Neg.** | **94%** | 0% | 63 | -0.0073 (0.0063) | | 8e-05 (3e-04) | |
| SVI 4—Housing Type & Transportation | Original | Insig. | 0% | **42%** | 31 | | 0.0039 (0.0018) | | 7e-03 (1e-02) |
| | Alpha | **Pos.** | 0% | **50%** | 12 | | 0.0039 (0.0012) | | 6e-03 (6e-03) |
| | Delta | **Pos.** | 0% | **54%** | 26 | | 0.0038 (0.0012) | | 2e-03 (3e-03) |
| | Omicron | **Pos.** | 6% | **70%** | 63 | -0.0012 (0.0004) | 0.0026 (0.0014) | 3e-03 (5e-03) | 1e-03 (5e-03) |

\* All provided p-values are statistically significant for $\alpha = 0.05$.

susceptibility to severe COVID-19 outcomes during the early stages of the pandemicm, reflecting the challenges faced by healthcare systems at the time (see S6 Fig, S1 and S2 Tables). However, this susceptibility diminished post the Alpha strain, likely due to targeted vaccination efforts prioritizing these groups.

**Table 6. Summary of key results for OMHR with SVI Level II.** For each variable and variant period, the overall association, the week % of statistical significance, the coefficients, and the p-values are provided. Those with statistical significance during more than half of the weeks are highlighted in bold.

| Variable | Variant | Assoc. | Weeks significant | | | Coeffs. avg (std) | | P-value* avg (std) | |
|---|---|---|---|---|---|---|---|---|---|
| | | | - | + | Total | - | + | - | + |
| SVI 1—Socioeconomic Status | Original | Mixed | 32% | 29% | 31 | -0.0017 (0.0006) | 0.0008 (0.0004) | 7e-03 (1e-02) | 0.02 (0.02) |
| | Alpha | **Pos.** | 0% | **75%** | 12 | | 0.0008 (0.0001) | | 8e-03 (1e-02) |
| | Delta | **Pos.** | 0% | **77%** | 26 | | 0.0016 (0.0009) | | 2e-03 (7e-03) |
| | Omicron | **Pos.** | 0% | **65%** | 63 | | 0.0018 (0.0023) | | 4e-03 (7e-03) |
| SVI 2—Household Characteristics | Original | **Pos.** | 0% | **65%** | 31 | | 0.0011 (0.0004) | | 3e-03 (7e-03) |
| | Alpha | **Pos.** | 0% | **75%** | 12 | | 0.0010 (0.0004) | | 4e-04 (5e-04) |
| | Delta | **Pos.** | 0% | **65%** | 26 | | 0.0005 (0.0004) | | 0.02 (0.01) |
| | Omicron | **Pos.** | 0% | **76%** | 63 | | 0.0009 (0.0012) | | 9e-03 (1e-02) |
| SVI 3—Racial & Ethnic Minority Status | Original | Mixed | 26% | 26% | 31 | -0.0011 (0.0004) | 0.0008 (0.0002) | 8e-03 (1e-02) | 2e-03 (3e-03) |
| | Alpha | Insig. | 0% | 0% | 12 | | | | |
| | Delta | Insig. | 27% | 0% | 26 | -0.0010 (0.0009) | | 1e-02 (2e-02) | |
| | Omicron | **Neg.** | **67%** | 0% | 63 | -0.0010 (0.0010) | | 4e-03 (1e-02) | |
| SVI 4—Housing Type & Transportation | Original | Insig. | 16% | 0% | 31 | -0.0011 (0.0004) | | 0.01 (0.01) | |
| | Alpha | Insig. | 0% | 0% | 12 | | | | |
| | Delta | Mixed | 31% | 27% | 26 | -0.0009 (0.0003) | 0.0004 (0.0002) | 0.01 (0.02) | 7e-03 (1e-02) |
| | Omicron | Insig. | 16% | 35% | 63 | -0.0031 (0.0015) | 0.0003 (0.0001) | 4e-04 (6e-04) | 0.01 (0.01) |

\* All provided p-values are statistically significant for $\alpha = 0.05$.

In contrast, Civilians with a Disability displayed increasing positive associations with both OMHR and CFR starting from the original strain, indicating ongoing challenges for this demographic. Single Parent Households also consistently showed positive associations with both mortality outcomes, though with variations in significance across different pandemic phases. Factors like financial strain, limited childcare, and barriers to healthcare may have exacerbated their risks. Notably, English Language Proficiency was predominantly negatively associated with both mortality outcomes, highlighting the diverse impacts of household characteristics on pandemic experiences. These findings underscore the need for tailored public health interventions that address the specific vulnerabilities of different household groups to better manage the impacts of the pandemic.

**Racial & ethnic minority status (SVI theme 3).** This theme exhibited a notable shift from positive to predominantly negative correlations with mortality outcomes from late 2020, indicating changes in risk factors or the effectiveness of mitigation strategies within these communities (Fig 7). The initial positive correlations during the early pandemic stages underscored the heightened vulnerability of racial and ethnic minorities to COVID-19. Similar associations are observed in the deeper analyses of Level III SVI factors (see S7 Fig, S1 and S2 Tables), with sporadic divergences that coincide with major pandemic waves. It is important to acknowledge that the impact of racial disparities may not be fully captured at the county level, and a more granular examination at the neighborhood level might provide additional insights into these associations.

**Housing type & transportation (SVI theme 4).** This factor showed a largely positive association with CFR, particularly after the emergence of the Alpha variant, suggesting that disparities in housing and transportation contributed to more severe COVID-19 outcomes. However, the relationships with OMHR were less consistent, pointing to the complex interplay between these factors and COVID-19 transmission dynamics (Fig 7, Tables 5 and 6). Despite these variances, the overall positive link between SVI theme 4 and CFR implies that disparities in housing and transportation may contribute to more severe COVID-19 outcomes, potentially due to delayed healthcare access or inadequate living conditions. However, Level III analysis (see S8 Fig, S1 and S2 Tables) revealed more variability and less significant associations among the individual variables of this theme with COVID-19 mortality, suggesting evolving dynamics through different pandemic stages. For example, the No Vehicle variable demonstrated a predominantly negative association with OMHR post-original strain, while Group Quarters showed a shift from a positive to a mainly negative association with OMHR as the pandemic progressed. These findings highlight the need for further investigation to clarify the underlying mechanisms and inform targeted interventions.

**Non-SVI explanatory variables.** The associations of non-SVI variables generally mirrored those seen in Level I, albeit with some variations in significance. Notably, the 65+ Age Group exhibited more periods of insignificance, likely due to its inclusion in SVI theme 2. Additionally, vaccination coverage trends indicated a protective effect against severe outcomes during the late Omicron phase, showing more positive periods compared to earlier in the pandemic. For conciseness, we are only showing the coefficients for SVI variables in Fig 7, while the coefficients for non-SVI variables are presented in the supporting information (see S10 Fig).

**Limitations and future directions.** Finally, we note that our study on the associations between disparities and COVID-19 mortality outcomes on all levels is subject to certain limitations. Potential confounders such as variations in case numbers, infection surges, and differences in hospital quality across regions could bias our findings, despite our efforts to control for them. Additionally, collinearity between explanatory variables, despite being checked with VIF scores, could inflate standard errors, complicating the interpretation of each variable's unique contributions. Our regression model's assumption of linearity may not fully capture the

complexities of these relationships, potentially affecting the accuracy of our estimates. The observed associations are complex and influenced by multiple underlying factors, and they do not necessarily imply causality. The county-level granularity of our data may not adequately reflect local variations that could impact these associations, and the coarse disaggregation of beds and hospitalization data may obscure some patterns. Furthermore, the rapidly evolving nature of the pandemic could introduce errors in data collection and reporting, affecting the accuracy and completeness of our datasets. Despite efforts to address these limitations, they should be carefully considered when interpreting our results. Further research and methodological enhancements are needed to provide more detailed and robust insights into these issues.

## Conclusion

This study offers valuable insights into the multifaceted relationships between disparities and COVID-19 mortality outcomes across U.S. counties throughout the pandemic timeline, utilizing indicators such as the Case Fatality Rate and Overall Mortality to Hospitalization Rate. The evidence highlights persistent correlations between healthcare access attributes, vaccination coverage, social vulnerability indices, rural-urban status, and COVID-19 mortality measures. The complex interplay between the various socioeconomic, demographic, and environmental factors captured by the SVI highlights the multidimensional nature of COVID-19 disparities. While certain associations persisted throughout the pandemic, such as the vulnerability of low socioeconomic status communities, others changed over time, influenced by public health interventions, evolving risks, and transmission dynamics. This underscores the importance of a nuanced understanding of how these disparities manifest and evolve, enabling targeted and adaptive strategies to mitigate the disproportionate burden on vulnerable populations during public health emergencies. Ongoing monitoring and analysis of these factors at multiple geographic scales will be crucial in informing equitable preparedness and response efforts for future pandemics. These findings underscore the critical need for holistic strategies that promote equitable access to healthcare resources, effective vaccination campaigns, and tailored interventions for disadvantaged groups.

Our findings contribute to expanding knowledge on how various disparities intersect to shape healthcare outcomes in a pandemic over time. These insights can inform evidence-based policies and resource allocation to alleviate disparities, enhancing preparedness against future pandemic threats. They also highlight the importance of continuous improvements in data collection, reporting systems, and methodologies to derive nuanced understandings. Overall, this study underscores the necessity of multifaceted and adaptive strategies grounded in health equity to create resilient communities in the face of public health crises. Future research should focus on more granular analyses at the local and community levels, employing qualitative and mixed-methods approaches to uncover details obscured by broader data. Further, exploring the interactions between various disparities through intersectional lenses and advancing methodologies to refine analyses remains crucial. Longitudinal studies assessing how these associations evolve can provide invaluable information for crafting policies that promote health equity and build resilient communities against public health crises.

## Supporting information

**S1 Fig. Time series of county exclusion percentages.** Percentage of excluded counties in each run of the analyses using the three times the mean of Cook's distance threshold.
(TIFF)

**S2 Fig. Correlation matrix of independent variables in Level I analysis.** Pearson correlation coefficients among independent variables for the level I analysis.
(TIF)

**S3 Fig. Correlation matrix of independent variables in Level II analysis.** Pearson correlation coefficients among independent variables for the level II analysis.
(TIF)

**S4 Fig. Correlation matrix of independent variables in Level III analysis.** Pearson correlation coefficients among independent variables for the level III analysis.
(TIF)

**S5 Fig. Temporal associations between COVID-19 outcomes and SVI socioeconomic status variables in Level III analysis.** SVI theme I is almost consistently positively associated with mortality measures and education and poverty amplify this association but the other measures provide less consistent relationships.
(TIF)

**S6 Fig. Temporal associations between COVID-19 outcomes and SVI household characteristics variables in Level III analysis.** Single-parent households show a consistent positive association with both CFR and OMHR.
(TIF)

**S7 Fig. Temporal associations between COVID-19 outcomes and SVI racial & ethnic minority status variables in Level III analysis.**
(TIF)

**S8 Fig. Temporal associations between COVID-19 outcomes and SVI housing type & transportation variables in Level III analysis.** The housing and transportation variables exhibited mixed patterns, predominantly inconsistent negative associations with COVID-19 mortality outcomes.
(TIF)

**S9 Fig. Time series of $R^2$ values for each SVI level analyses.**
(TIFF)

**S10 Fig. Temporal associations between COVID-19 outcomes and non-SVI theme variables in Level II analysis.** The associations here are primarily similar to those of Level I analysis (see Fig 5), with lower levels of significance in some cases. The 65+ Age Group specifically exhibits more frequent insignificant periods, which is expected as SVI theme 2 has it as one of its constituent variables. Vaccination coverage also exhibits more positive periods concerning CFR in the late Omicron phase compared to Level I analyses. SVI variables for this analysis are presented in the main body of the paper, see Fig 7.
(TIF)

**S1 Table. Summary of key results for CFR with SVI Level III variables.** For each variable and variant period, the overall association, the week % of statistical significance, the coefficients, and the p-values are provided. Those with statistical significance during more than half of the weeks are highlighted in bold.
(PDF)

**S2 Table. Summary of key results for OMHR with SVI Level III variables.** For each variable and variant period, the overall association, the week % of statistical significance, the coefficients, and the p-values are provided. Those with statistical significance during more than half

of the weeks are highlighted in bold.
(PDF)

**S3 Table. $R^2$ values for each SVI level analyses.**
(PDF)

**S4 Table. Summary of key results for CFR with non-SVI explanatory variables in the Level II analysis.** The associations between SVI theme variables and CFR are presented in Fig 7. For each variable and variant period, the overall association, the week % of statistical significance, the coefficients, and the p-values are provided.
(PDF)

**S5 Table. Summary of key results for OMHR with non-SVI explanatory variables in the Level II analysis.** The associations between SVI theme variables and OMHR are presented in Fig 7. For each variable and variant period, the overall association, the week % of statistical significance, the coefficients, and the p-values are provided.
(PDF)

## Author Contributions

**Conceptualization:** Fardin Ganjkhanloo, Farzin Ahmadi, Kimia Ghobadi.

**Data curation:** Fardin Ganjkhanloo, Ensheng Dong, Felix Parker.

**Formal analysis:** Fardin Ganjkhanloo, Farzin Ahmadi, Kimia Ghobadi.

**Funding acquisition:** Kimia Ghobadi.

**Methodology:** Fardin Ganjkhanloo, Farzin Ahmadi, Kimia Ghobadi.

**Project administration:** Fardin Ganjkhanloo, Kimia Ghobadi.

**Resources:** Kimia Ghobadi.

**Software:** Fardin Ganjkhanloo.

**Supervision:** Kimia Ghobadi.

**Validation:** Fardin Ganjkhanloo, Kimia Ghobadi.

**Visualization:** Fardin Ganjkhanloo.

**Writing – original draft:** Fardin Ganjkhanloo, Farzin Ahmadi.

**Writing – review & editing:** Fardin Ganjkhanloo, Farzin Ahmadi, Ensheng Dong, Felix Parker, Lauren Gardner, Kimia Ghobadi.

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
