## [Decision Letter · Decision Letter 0]

19 Jul 2024

Evolving Patterns of COVID-19 Mortality in US Counties: A Longitudinal Study of Healthcare, Socioeconomic, and Vaccination Associations

PGPH-D-24-00955

Dear Prof. Ghobadi,

We are pleased to inform you that your manuscript 'Evolving Patterns of COVID-19 Mortality in US Counties: A Longitudinal Study of Healthcare, Socioeconomic, and Vaccination Associations' has been provisionally accepted for publication in PLOS Global Public Health.

Best regards,

Gautam I Menon, PhD

Academic Editor

Reviewer Comments (if any, and for reference):

Reviewer's Responses to Questions

**Comments to the Author**

1. Does this manuscript meet PLOS Global Public Health’s publication criteria? Is the manuscript technically sound, and do the data support the conclusions? The manuscript must describe methodologically and ethically rigorous research with conclusions that are appropriately drawn based on the data presented.

Reviewer #1: Yes

Reviewer #2: Yes

2. Has the statistical analysis been performed appropriately and rigorously?

Reviewer #1: Yes

Reviewer #2: No

3. Have the authors made all data underlying the findings in their manuscript fully available (please refer to the Data Availability Statement at the start of the manuscript PDF file)?

Reviewer #1: Yes

Reviewer #2: Yes

4. Is the manuscript presented in an intelligible fashion and written in standard English?

Reviewer #1: Yes

Reviewer #2: Yes

5. Review Comments to the Author

Reviewer #1: I would like to recommend the authors for a rigorous study and a well summarized study.

Overall, the article effectively introduces the context of the study by emphasizing the significance of pandemic preparedness strategies and the impact of the COVID-19 pandemic in the United States. It clearly states the research objective and the specific variables of interest, which are described in detail.

The description of the methodology provides a clear framework for understanding how the associations between factors and COVID-19 mortality outcomes were examined, allowing for potential replication in other geographical locations, with or without modifications.

Despite the limitations, the study presents key findings, highlighting both persistent and dynamic correlations between various factors and COVID-19 mortality measures. These findings are highly valuable to the subject under consideration.

It is noteworthy that the protective effects of access to health resources, including hospital beds and higher vaccination coverage, as well as the association between higher Social Vulnerability Index and worse health outcomes, are well known. Therefore, the emphasis on persistent and dynamic correlations between these factors and COVID-19 mortality measures not only adds value to the subject but also underscores its significance for improving health and well-being.

Furthermore, the article emphasizes the importance of implementing targeted policies and interventions to address the disparities identified in marginalized communities. It highlights the urgent need for such interventions to improve public health outcomes and effectively combat future pandemics.

Reviewer #2: This topic is crucial, offering valuable and multifaceted insights that can inform pandemic interventions. The research questions are well-defined, and the methods are detailed, comprehensive, and aligned with the research objectives. The results and discussion are thoroughly elaborated and precise. Overall, the reading was enjoyable.

6. PLOS authors have the option to publish the peer review history of their article (what does this mean?). If published, this will include your full peer review and any attached files.

**Do you want your identity to be public for this peer review?** For information about this choice, including consent withdrawal, please see our Privacy Policy.

Reviewer #1: No

Reviewer #2: No
